# Exploring burnout, perfectionism, and moral injury among UK physiotherapists: A qualitative study on professional fulfilment and well-being

Glykeria Skamagki[1], Laura Blackburn[2], Daniel Biggs[3], Maria Kolitsida[1], Cameron Black[4], Sivaramkumar Shanmugam[2]*

1 School of Sport, Exercise and Rehabilitation Sciences, College of Life and Environmental Sciences, University of Birmingham, Birmingham, United Kingdom, 2 Department of Physiotherapy and Paramedicine, School of Health and Life Sciences, Glasgow Caledonian University, Glasgow, United Kingdom, 3 Academy of Sport and Wellbeing, University of Highlands and Islands, UHI Perth, Perth, United Kingdom, 4 Division of Occupational Health, Buckinghamshire Healthcare NHS Trust, Aylesbury, United Kingdom

* Sivaram.shanmugam@gcu.ac.uk

**Data Availability Statement:** All relevant data are within the manuscript and its Supporting information files.

## Abstract

### Background

Burnout, perfectionism, and moral injury are prevalent issues among healthcare professionals, including physiotherapists. The demanding nature of the profession, compounded by high workloads, emotional and physical exhaustion, and systemic challenges, has significant implications for the well-being and professional satisfaction of physiotherapists. This study aimed to explore these issues, by providing a qualitative exploration of UK physiotherapists' lived experiences.

### Objectives

To explore the lived experiences of UK physiotherapists regarding burnout, perfectionism, and moral injury, and to develop a comprehensive understanding of the personal and professional impacts of these issues to inform the development of effective support systems and interventions.

### Methods

This qualitative exploratory study involved semi-structured interviews with 12 UK physiotherapists. Framework approach was used to identify key themes and patterns in the data, providing a nuanced understanding of the challenges faced by physiotherapists.

### Results

Four primary themes emerged: (1) Physiotherapy Under Pressure: Workload, Burnout, and Perfectionism, (2) Interpersonal Dynamics and Support Systems, (3) Professional Fulfilment and Identity, and (4) Work-Life Balance and Well-being. Physiotherapists reported high levels of burnout and exhaustion due to relentless workloads, exacerbated by and after

**Funding:** The author(s) received no specific funding for this work.

**Competing interests:** The authors have declared that no competing interests exist.

the COVID-19 pandemic. Perfectionism further contributed to emotional exhaustion and feelings of inadequacy. Inconsistent management support, bureaucratic challenges, and a lack of career progression opportunities were significant stressors. Effective team dynamics and support systems were crucial in mitigating stress, yet many faced interpersonal challenges such as criticism and bullying. Achieving work-life balance was a persistent struggle, highlighting the need for organisational changes to support flexibility and well-being.

## Conclusion

Burnout, perfectionism, and moral injury significantly impact physiotherapists' well-being and professional satisfaction. Addressing these issues requires systemic changes within healthcare organisations to provide robust support systems, flexible working conditions, and opportunities for professional development.

## Introduction

Burnout is increasingly prevalent among physiotherapists and has been described as a critical issue with significant implications for both healthcare professionals and the quality of patient care [1]. Burnout is characterised by emotional exhaustion, depersonalisation, and a reduced sense of personal accomplishment [2]. Recent studies [1–4] have highlighted the growing impact of burnout in this profession as for example, Biggs et al. [3] revealed that 96% of UK physiotherapists experience moderate to high levels of burnout, with perfectionism and moral injury being significant predictors. This study underscores the urgent need for targeted interventions to address burnout and its associated factors.

Although burnout is not a new concept, the COVID-19 pandemic has further exacerbated these challenges, leading to increased workload and stress among physiotherapists. Recent systematic reviews and meta-analyses have shown a significant rise in burnout levels during the pandemic. Specific factors contributing to burnout during the COVID-19 pandemic include heightened patient loads, significant staff shortages, and exacerbated mental health challenges faced by healthcare workers. For example, Clarke et al. [5] found that 40% of musculoskeletal allied health practitioners, including physiotherapists, reported high levels of emotional exhaustion, with many attributing this to the increased demands and stress brought on by the pandemic. Similarly, Venturini et al. [1] reported a global prevalence of burnout at 8%, highlighting the widespread impact of this issue.

Burnout in physiotherapy has profound implications not only for the individuals affected by it but also for the healthcare system as a whole. It can lead to decreased job satisfaction, increased absenteeism, and higher turnover rates, which in turn affect patient care and outcomes [2, 6]. Additionally, burnout can result in significant mental health issues such as anxiety, depression, and substance abuse [7]. These findings align with the broader literature on healthcare professional burnout, which consistently highlights the detrimental effects on both personal well-being and professional performance [8].

Despite the substantial quantitative data on burnout prevalence and risk factors, there is a gap in the literature regarding the qualitative experiences of physiotherapists, particularly in the UK. Quantitative research has created a foundation by identifying key factors such as perfectionism and moral injury [3, 9] but did not delve into the personal and emotional experiences of physiotherapists. Exploring the perceptions and experiences of physiotherapists who

have been impacted from burnout is crucial as it allows for a deeper understanding of the nuances and complexities of this critical issue in healthcare. It also provides insights into how burnout manifests in daily practice, affects professional identity, and impacts personal well-being.

Given the high prevalence of burnout and its severe implications, this research is timely and essential for advancing our understanding and support of physiotherapists in the UK and beyond. Listening to physiotherapists' voices about their experiences and perspectives is crucial for developing a comprehensive understanding of burnout and the strategies they employ to manage it. By exploring these personal narratives, we can gain insights into the real-world challenges faced by physiotherapists and the coping mechanisms that may not be evident in quantitative studies alone.

The aim of this study is to explore the lived experiences of UK physiotherapists regarding burnout, perfectionism, and moral injury and to develop a comprehensive understanding of the personal and professional impacts of these issues and to inform the development of effective support systems and interventions. Although qualitative research cannot directly inform policies, it can contribute by providing in-depth insights into the lived experiences of physiotherapists. These findings may be used to support government initiatives, such as the NHS People Plan, by emphasising the importance of mental health resources, tailored training programs, and workplace flexibility to mitigate burnout.

## Methods

### Design

This study employed a generic qualitative approach to explore and understand the experiences of UK physiotherapists regarding burnout, perfectionism, and moral injury. A generic qualitative approach is well-suited for investigating phenomena or processes without aligning with a specific qualitative methodology, allowing for a broad and flexible exploration of clinical issues [10]. This approach enabled researchers to gather rich, detailed narratives that offer deeper insights into the complexities of burnout and related factors. The methods utilised in this study involved semi-structured interviews, which are highly effective for obtaining detailed insights into participants' experiences. This format provided the necessary flexibility to delve into physiotherapists' perspectives while maintaining enough structure to ensure all key topics are addressed.

Ethical approval for the study was granted by the Glasgow Caledonian University (HLS/PSWAHS/22/017), ensuring that all research activities were conducted in accordance with ethical standards. The study was also guided by the Consolidated Criteria for Reporting Qualitative Research (COREQ) checklist, which provided a comprehensive framework for ensuring rigour and transparency in qualitative research.

### Participant recruitment

To follow up on the "Physiotherapy Under Pressure" study conducted by Biggs et al. [3] participants were recruited using purposive sampling. This approach involved selecting physiotherapists who had previously completed the survey and had expressed interest in participating in a qualitative follow-up study. Inclusion criteria included physiotherapists working in the UK, being over 21 years of age, working full-time or part-time and having experienced burnout. Exclusion criteria applied to those being retired or a student. All potential participants were contacted via email and invited to participate in the study. Those who expressed interest were provided with detailed information about the study and asked to consent to participate. Participants were informed that consent was an ongoing process and that they could withdraw at

any time. This information was reiterated when the participant information sheet was emailed during the recruitment process for the qualitative study. Written consent was sought at this stage. During the interview, verbal consent was sought again at the outset and was documented by the researcher.

## Data collection

Recruitment emails were sent out in October 2023 through to the end of January 2024, and data were collected through semi-structured interviews conducted online via Zoom between November 2023 and March 2024. Each interview lasted between 45 to 60 minutes, allowing sufficient time to explore participants' experiences in depth. The interview guide was developed based on a review of relevant literature and refined through feedback from the research team and pilot interviews. A pilot study was initially conducted with one participant, followed by a debrief session where the qualitative researcher made notes for changes. Feedback from the pilot interview led to minor adjustments in the wording and sequencing of questions to improve clarity and flow. A second pilot study confirmed the final interview guide, as no further changes were needed.

The interview guide consisted of open-ended questions designed to elicit detailed narratives about participants' experiences with burnout, perfectionism, and moral injury within their physiotherapy roles. Examples of questions included: "Can you describe how burnout has impacted your professional performance?" and "What support, if any, have you received from your workplace to manage these challenges?" This approach ensured comprehensive coverage of relevant topics while allowing participants to express their views and experiences freely. During the interviews, the researcher took notes on body language, nonverbal responses, and the use of metaphors to capture the full depth of participants' experiences. Interviews continued until data saturation was achieved, meaning no new themes or insights were emerging from the data [11]. This approach ensured a comprehensive understanding of the phenomena being studied.

To ensure the reliability and accuracy of the data, all interviews were recorded and transcribed verbatim using Zoom. The transcripts were then meticulously reviewed and edited by the qualitative researcher for accuracy. Data were stored securely in an encrypted space on OneDrive, and all identifying information was removed to maintain participant anonymity. Each participant was assigned a pseudonym (e.g., Participant 1, Participant 2) for the reporting process. Participants were informed of their right to withdraw from the study at any time. They were encouraged to speak openly and were given the option to take breaks or skip questions due to the sensitive nature of the topic. After the interviews, the lead researcher took time to reflect and documented detailed notes on the conversation and the significant points raised by the participants. This additional context enriched the data, providing a deeper insight into the participants' experiences.

## Data analysis

The data analysis followed a framework approach developed by Spencer et al. [12], comprising five intertwined stages. The analytic process began with the first author familiarising herself with the data by reading and re-reading the transcripts to immerse herself in the content. This was followed by the initial coding of the data. The qualitative researcher gathered all similar data into clusters and created initial categories. In total, 220 codes were created from the 12 interviews, and a coding matrix was created in NVivo 14 for each of them. Identifying codes and clustering them into categories was aided by mapping exercises, which helped visualise the relationships between different codes. Descriptions of each category were created to capture

the essence of the data, and categories with similar meanings were then further grouped under broader thematic titles. The labels and descriptions used for the raw data were revisited and carefully examined to ensure they accurately reflected the interview data.

During this iterative process, coded content was reviewed and, if necessary, moved or created a combined different category. This thorough process generated nine final categories, which were then further refined into four analytical overarching themes. To enhance the rigour and transparency of the findings, a reflective diary was maintained throughout the research process. This diary enabled the documentation of personal biases, assumptions, and reflections, acknowledging how the researchers' background and experiences might influence the interpretation. The final thematic structure provided a comprehensive understanding of the experiences of UK physiotherapists with burnout, perfectionism, and moral injury, contributing valuable insights to the survey results [3] and potentially inform the development of interventions and support strategies.

## Results

### Participants characteristics

Twelve Physiotherapists took part in the study, including 8 females and 4 males, from various specialties and organisational types. The sample size of 12 participants was determined by data saturation, where no new themes emerged from the interviews. Participants were purposively selected based on their completion of a prior survey and willingness to participate in qualitative follow-up, ensuring alignment with the study objectives. Data collected included age, gender, area of physiotherapy, organisation type and length of service (Table 1). This diverse sample provided a broad perspective on the experiences of physiotherapists across different contexts.

Findings in this study are structured around four overarching themes: Physiotherapy Under Pressure: Workload, Burnout, and Perfectionism, Interpersonal Dynamics and Support Systems, Professional Fulfilment and Identity and Work-Life Balance and Well-being.

### Theme 1: Physiotherapy under pressure: Workload, burnout, and perfectionism

This theme explored the complex relationship between workload, burnout, and perfectionism among UK physiotherapists. Participants' narratives offered a detailed insight into the

**Table 1. Participants characteristics.**

| ID | Age | Gender | Area of Practice | Organisation Type | Banding Type |
|---|---|---|---|---|---|
| Participant 1 | 32 | Female | Musculoskeletal | NHS Trust | Band 6 |
| Participant 2 | 45 | Male | Musculoskeletal | Private Practice | Band 7 |
| Participant 3 | 29 | Female | Musculoskeletal | Private Practice | Band 6 |
| Participant 4 | 34 | Female | Paediatrics | NHS Trust | Band 6 |
| Participant 5 | 50 | Male | First Contact Practitioner | GP surgery | Band 8b |
| Participant 6 | 38 | Female | Cardiorespiratory | NHS Trust | Band 6 |
| Participant 7 | 31 | Female | Orthopaedics | NHS Trust | Band 6 |
| Participant 8 | 46 | Male | Community | NHS Trust | Band 6 |
| Participant 9 | 38 | Female | First Contact Practitioner | Private Practice | Band 8a |
| Participant 10 | 42 | Male | Community | NHS Trust | Band 6 |
| Participant 11 | 35 | Female | Mental Health | NHS Trust | Band 7 |
| Participant 12 | 26 | Female | Rotation | NHS Trust | Band 5 |

pervasive issue of burnout, highlighting not just the immediate effects of heavy workloads, but also the long-term implications for professional fulfilment and personal well-being.

**Burnout and exhaustion.**  The high workloads faced by physiotherapists had severe impacts on their well-being, leading to burnout and exhaustion. Participants frequently reported emotional and physical exhaustion due to the relentless pace of their work. They often managed numerous patients daily, with back-to-back appointments and minimal breaks in between. This relentless schedule left little time for recovery, leading to physical exhaustion. "*The back-to-back patient appointments with no time to breathe in between have pushed me to my limits,*" shared one physiotherapist.

This lack of recovery time resulted in feeling constantly overwhelmed and struggling to maintain their professional responsibilities. One participant highlighted the emotional toll of the job, stating, "*There have been times when I felt so drained, both physically and emotionally, that I could barely function.*" This emotional exhaustion often led to decreased motivation impacting their ability to provide quality patient care as the role requires to continuously engage with patients, listen to their concerns and provide emotional support.

**The aftermath of COVID-19.**  The COVID-19 pandemic exacerbated the challenges faced by physiotherapists, significantly impacting their workloads, mental health, and the support they received from management. Challenges included inadequate staffing, limited resources, and increased administrative demands, all of which significantly contributed to burnout. This period brought an increase in both anxiety and workload, as well as the need for significant organisational changes. Participants reported handling more patients than ever, often under stressful conditions. "*The heightened anxiety affected my mental health and work performance, making it challenging to maintain professional responsibilities,*" noted one interviewee, indicating how the already high demands of their job had been intensified. This increased pressure contributed significantly to burnout, as physiotherapists found themselves stretched thin, both physically and mentally.

The uncertainty and stress related to COVID-19 continued at the end of the pandemic. Staffing challenges also became more pronounced during and after this era. Teams were left short-staffed which further increased the burden on those who remained. One participant shared, "*We were already dealing with high workloads, but with colleagues off sick or quitting, it became nearly impossible to manage.*"

The transition to remote work introduced a new set of challenges. Physiotherapists had to adapt quickly to virtual consultations and new technologies, often without adequate support from management. "*The lack of consistent support from management during COVID-19 has made the situation even more stressful and frustrating,*" noted one physiotherapist.

However, remote work remained after the pandemic and increased administrative burdens. Many physiotherapists found themselves dealing with more paperwork and bureaucratic processes than before which many felt they had not "signed up" to do. As explained by one participant, "*I would spend hours on admin tasks, and then come back to the clinic the next day only to find we were behind on patient care tasks. It felt like we were constantly playing catch-up.*" The need to juggle multiple roles left many feeling they were unable to give their best to their patients, further contributing to feelings of professional dissatisfaction and burnout.

**Work environment.**  The work environment emerged as a critical factor influencing the burnout and resilience of physiotherapists. Inconsistent support from management was significantly affecting physiotherapists' ability to manage their workloads and maintain their motivation. Many participants expressed also feelings of isolation and frustration. One physiotherapist said, "*The lack of consistent support from management makes it hard to stay motivated.*"

The absence of consistent management support was not only demoralising but also led to a sense of helplessness exacerbating the already high levels of stress associated with their role. Physiotherapists frequently found themselves navigating complex and demanding situations without the necessary guidance or backing from their managers. This lack of support contributed to a feeling of being undervalued and increased the risk of burnout. Another participant noted, "*When management is not there, you start to question why you are putting in all this effort.*"

Bureaucratic challenges were frequently cited as significant sources of stress for physiotherapists. The changes in protocols, coupled with the increasing administrative workload, added layers of complexity to their daily routines. "*The bureaucracy and constant changes in protocols are exhausting,*" shared one physiotherapist. These hurdles not only increased the time spent on non-clinical tasks but also detracted from the time available for patient care, which many physiotherapists found particularly frustrating. They often felt disconnected from the practical realities of clinical work, leading to inefficiencies and added stress. One physiotherapist explained, "*I spend hours on paperwork that seems pointless, and it takes away from the time I should be spending with patients.*"

Participants called for better support systems as the current structures were often seen as rigid and unsupportive, contributing to the high levels of stress and burnout. Suggestions included streamlined processes for handling administrative tasks, better communication from management, and the introduction of more flexible work arrangements. One participant explained, "*The system needs an overhaul. We need structures that support us, not ones that add to our stress.*"

Finally, the impact of workload and burnout varied significantly across different career stages, highlighting the unique challenges faced by physiotherapists at various points in their professional journeys. Newly graduated physiotherapists often felt immense pressure and lacked sufficient support. One participant shared a distressing experience: "*I used to lock myself in the storage room to cry because I felt so much pressure, and my seniors didn't have the time to help me—they only criticised.*" Similarly, senior physiotherapists also faced considerable challenges, often feeling overwhelmed by their responsibilities. These included supervising students, managing patients, provide training for others and handling bureaucratic tasks.

**Perfectionism.** Perfectionism was another critical factor that influenced the burnout and well-being of physiotherapists. The high standards and expectations that physiotherapists set for themselves often led to increased stress and emotional exhaustion. They explained that felt inadequate and constantly strived to meet unattainable standards. "*I always feel like I need to do everything perfectly, and it just wears me out,*" shared one participant. This pursuit of perfection not only contributed to emotional exhaustion but also affected their professional performance and personal life. One participant noted, "*The constant need to be perfect in everything I do is exhausting. It feels like I'm never good enough, no matter how hard I try. I just want to quit from life*".

Some participants employed various strategies to cope with perfectionism, including therapy, self-awareness practices, and developing self-compassion. However, other challenges highlighting the difficulties in managing their high standards and expectations. "*Sometimes, despite my efforts, I still find it hard to let go of my standards,*" shared one physiotherapist.

**Moral injury.** Physiotherapists often entered the profession with a strong sense of duty and commitment to providing high-quality patient care. However, participants discussed that systemic issues such as inadequate resources, high workloads, and bureaucratic obstacles frequently prevent them from fulfilling their roles causing deep frustration. One participant explained: "*It's incredibly frustrating when I know what the best care for my patient would be, but I skip it* ".

In addition, the ongoing struggle to reconcile professional values with the constraints of the healthcare system often resulted in significant moral distress. Participants found themselves in situations where they had to compromise on the quality of care due to factors beyond their control, such as limited time, insufficient staffing, or restrictive protocols. This led to feelings of guilt, shame, and dissatisfaction. For example, one participant said, "*I feel like I'm constantly fighting against a system that doesn't care about the patients or staff*,".

The emotional and psychological consequences of moral injury were clear as physiotherapists reported feelings of anger, betrayal, and disillusionment with the healthcare system leading to burnout, and even a desire to leave the profession. One participant explained "*I feel like I'm constantly battling the system. It's draining and makes me question why I chose this profession. And imagine that I was a mature student and this was my dream job*".

## Theme 2: Interpersonal dynamics and support systems

This theme highlights the critical role that interpersonal relationships and support systems play in the professional and personal lives of physiotherapists with burnout. The narratives highlight the impact of team dynamics, support networks, interpersonal challenges, and the emotional repercussions on the well-being and job satisfaction of physiotherapists.

**Team dynamics.**   Team dynamics were crucial in shaping the work environment and overall experience of participants. They explained that effective team support and interaction significantly mitigates stress and burnout. Many participants emphasised the importance of having a cohesive team where members understand and support each other. One participant noted, "*Having a supportive team makes a huge difference. We look out for each other and help manage the workload.*"

However, not all participants experienced positive team dynamics. Some reported a lack of understanding and support from their teams and managers, which exacerbated their stress levels. "*The lack of consistent support from management makes it hard to stay motivated*," explained one interviewee. This inconsistency especially in managerial support left them feeling isolated and overwhelmed, highlighting the need for better leadership.

Effective communication within teams also emerged as a pivotal factor. Participants who reported good communication within their teams felt more equipped to handle their responsibilities and less isolated in their roles. "*Good communication within the team helps us stay on the same page and manage the day efficiently*," noted one physiotherapist.

**Support networks.**   Support networks, both within and outside the workplace, were essential for the emotional and professional well-being of physiotherapists. These were categorised as formal, such as management-led initiatives, and informal, such as peer support networks, to emphasise their complementary roles. Participants highlighted the benefits of having mentors or peers they could turn to for guidance. They explained that their colleagues often serve as a primary source of support, providing both emotional and professional assistance. "*Support from my colleagues has been invaluable. They understand the pressures of the job and are always there to lend a hand or offer advice*," shared one participant.

In addition, emotional support systems extended beyond the workplace, encompassing family and friends who provided a crucial safety net. "*Having support at home helps me cope with the stresses of the job. My family and friends are my rock*," explained another interviewee. This external support was vital in helping physiotherapists with burnout to manage the emotional toll of their work.

**Interpersonal challenges.**   Despite the benefits of strong support networks, many physiotherapists faced significant interpersonal challenges. Participants described constructive criticism as feedback aimed at improvement, whereas others faced bullying as comments involved

personal attacks undermining their professional competence."Being criticised and bullied by colleagues or managers was a common experience that negatively impacted their job satisfaction and mental health. "*I felt constantly criticised by my seniors, which made me doubt my abilities and increased my stress levels*," shared one participant. The impact of these challenges was particularly pronounced for newly graduated physiotherapists who often felt immense pressure and lack sufficient support.

Minimal team interaction and conflicts with colleagues further contributed to these challenges. "*There were times when I felt completely isolated because my colleagues were too busy to even talk to each other*" noted one physiotherapist. These interpersonal conflicts not only hinder team cohesion but also increase individual stress and frustration. Such negative interactions can create a toxic work environment, leading to further isolation and burnout.

**Emotional impact.**   Physiotherapists frequently reported that the lack of support and negative interactions led to feelings of isolation and emotional dysregulation. Emotional dysregulation often led participants to withdraw from their colleagues, exacerbating feelings of professional isolation and further contributing to burnout. "*The constant criticism and lack of support made me feel isolated and emotionally drained. I was wondering 'what am I doing here?'*"," explained one participant. This emotional toll affected their professional performance but also spilled over into their personal lives, impacting their overall well-being. Participants noted that the stress and emotional exhaustion from work often strained their relationships with family and friends. "*By the end of the day, I'm so exhausted that I can barely spend time with my family*" shared another physiotherapist.

## Theme 3: Professional fulfilment and identity

This theme highlighted the multifaceted nature of professional fulfilment and identity among UK physiotherapists. The narratives from the interviews revealed the complexities of finding professional satisfaction in a demanding and often undervalued profession.

**Recognition and appreciation.**   Recognition and appreciation from patients and colleagues were crucial for professional fulfilment among participants. The acknowledgment of their efforts provides a sense of validation and boosts morale. One participant expressed, "*It's been appreciated either you know by patients or sometimes by colleagues that you've been helpful.*" This appreciation not only reinforced their professional efforts but also fostered a sense of belonging within the professional community.

Appreciation in work also contributed to job satisfaction. Physiotherapists who felt valued in their roles reported higher levels of motivation and commitment. Being recognised for their contributions was not just a morale booster but a critical aspect of their professional identity. One interviewee shared, "*Recognition by patients and colleagues makes a big difference. It feels like all the hard work is acknowledged and appreciated. Then I know I am at the right place*".

**Sense of accomplishment.**   Physiotherapists derived significant satisfaction from their ability to make a difference in their patients' lives. This sense of accomplishment was closely tied to their confidence in their professional roles. One participant shared, "*Feeling useful and productive in your work helps maintain motivation and purpose. . .That I am in the right place and I can do this and I can do that..*" reflecting the intrinsic motivation and self-assurance that came from performing well in the job. Participants often spoke about the joy and fulfilment they experienced when their efforts led to positive patient outcomes. "*When a patient tells me that they feel better because of my treatment, it makes all the hard work worth it*," said one physiotherapist.

**Career development and growth.**   Many physiotherapists expressed that while continuous professional development (CPD) is essential for maintaining the requirements of their

professional body and enhancing their skills [13] the demands of their workload left them feeling overwhelmed and exhausted. One participant highlighted, "*Continuous learning is crucial for my growth and to feel confident in my role, but the expectations and time commitment can be daunting.*" This sentiment was echoed by others who felt that the time and effort required for CPD, on top of their already demanding workloads, contributed to their burnout.

Moreover, the expectations set by professional bodies for CPD often felt unrealistic and added to the sense of pressure. For example, professional bodies require physiotherapists to complete mandatory CPD hours annually and obtain certifications to maintain their registration. "*The professional bodies have set such high standards for CPD, and it feels like there's no flexibility. It just adds another layer of stress, and they are oblivious to what is happening to us and how demanding is our profession"* remarked another participant. The lack of structured and supportive CPD opportunities within their organisations also emerged as a concern. Many physiotherapists reported that they had to pursue CPD on their own time, which often encroached on their personal time and work-life balance. "*We're expected to do all this CPD, but there's no time allocated for it during work hours. It's like we're expected to do it in our own time, which just adds to the burnout,*" noted a participant.

Working outside of their roles to create opportunities was a strategy employed by some physiotherapists to overcome the lack of formal career progression pathways. They took on additional responsibilities or pursued further education to broaden their career prospects. "*I had to take extra courses on my own time to stay ahead,*" *shared one participant*". However, the absence of structured career progression opportunities often led to frustration and a feeling of being undervalued. Another participant articulated, "*The lack of career progression opportunities makes me feel stagnant. I need new challenges to stay motivated.*"

**Career stagnation and change.** Career stagnation was a significant issue, with many physiotherapists feeling stuck in the same role for extended periods. This lack of movement not only hindered their professional growth but also contributed to job dissatisfaction. Participants discussed the desire for career change as a natural response to this stagnation. Participants explored career options such as private practice, academic roles, or further specialisation to counter stagnation. Some contemplated shifting careers entirely to burnout and others to escape the monotony and seek new challenges. This desire was driven by the need for professional fulfilment and the aspiration to find a role that offered better growth prospects. One participant shared. "*I've been doing the same thing for years now, and it feels like I'm just spinning my wheels. I'm seriously thinking about switching careers because I need something new and challenging to keep me motivated and balanced.*"

Some physiotherapists considered shifting to different roles within healthcare or entirely different fields. "*I've thought about leaving physiotherapy altogether because I don't see any future growth here,*" shared one participant. This desire reflected the need for a more supportive workplace that offers opportunities for career advancement and work-life balance. "I've been doing the same thing for years, and it feels like there's no way out," noted one interviewee.

## Theme 4: Work-life balance and well-being

This theme captured the multifaceted aspects of work-life balance, coping mechanisms and the importance of well-being among physiotherapists. The insights provided by the participants highlight the critical need for systemic changes to support physiotherapists in achieving a healthier balance between their professional and personal lives.

**Work-life integration and flexibility.** Participants frequently highlighted the challenges of maintaining a healthy work-life balance. The demanding nature of their job often led to

poor work-life balance, with many struggling to separate their professional responsibilities from personal time. Some participants reported working an additional 10–15 hours per week due to increased administrative and clinical demands. "*It's hard to switch off after a long day at work. I often find myself thinking about patients and pending tasks even when I'm home.*" The struggle with maintaining work-life balance was evident as many physiotherapists found their personal lives impacted by their professional commitments. "*I used to have hobbies and interests outside work, but now I hardly find the time or energy for them,*" shared by another participant.

Flexibility in scheduling and the ability to set boundaries emerged as crucial factors for achieving better work-life integration. "*Being able to adjust my schedule and take breaks when needed has been a game-changer for my mental health,*" noted one physiotherapist. This flexibility (where possible) allowed them to manage their workload more effectively and allocate time for personal activities, thereby reducing stress.

However, this was not always supported by the organisational structures in place. Many participants expressed the need for better support systems and resources to help manage their workloads. "We need more support from the management to ensure that breaks are respected, even if this is for lunch, a walk or a counselling session" emphasised a participant, highlighting a gap in current practices.

**Personal well-being and coping mechanisms.** The toll on personal well-being was a recurrent issue, with many physiotherapists reporting mental health struggles, physical health symptoms, and sleep disturbances. "*The constant stress and pressure have led to frequent headaches and insomnia,*" shared one participant, highlighting the physical manifestations of work-related stress. Emotional resilience was identified as a key factor in managing these challenges. Developing self-compassion and seeking professional help, such as therapy, were common coping strategies. "*Therapy has been instrumental in helping me deal with the emotional burden of my job,*" noted one physiotherapist.

Exercise and self-awareness practices were also frequently mentioned as effective coping mechanisms. "*Regular exercise has been my go-to stress buster. It helps me clear my mind and stay physically fit,*" shared a participant. Another added, "*Practicing mindfulness and staying aware of my emotional state has been crucial in managing my stress levels.*" Despite these efforts, many physiotherapists still struggled to find a balance, indicating the need for more systemic support. "*No matter how many coping strategies I employ, the workload and pressure sometimes feel overwhelming,*" confessed one participant, highlighting the limitations of individual efforts in the absence of broader organisational changes.

**Management respecting work-life balance.** The role of management in respecting and supporting work-life balance was emphasised by many participants. The need for better mental health support and resilience-building initiatives from management was a recurring theme. "*We need management to acknowledge the stress we are under and provide resources and the time to support our mental health*" stated one physiotherapist.

Support from family and friends played a critical role in helping physiotherapists cope with their demanding jobs. "*My family has been my rock. Their support keeps me going,*" shared one participant, illustrating the importance of a strong personal support network. However, there was a clear call for more structured support from the workplace to complement these personal networks. Participants also expressed a need for management change and training to foster a more supportive work environment. This included providing more resources, improving communication, and creating a culture that values and respects work-life balance. "*Management needs to listen to our concerns, recognise burnout and take concrete steps to reduce our workload and stress,*" emphasised by one physiotherapist.

## Discussion

This study provides an in-depth exploration of the factors contributing to burnout among UK physiotherapists. The narratives revealed the intricate interplay between workload, burnout, perfectionism, and the professional identity of physiotherapists, offering critical insights into the systemic issues affecting their well-being and coping strategies as well.

Burnout and workload are significant, interconnected issues. The high demands placed on physiotherapists often lead to severe emotional and physical exhaustion. This aligns with the broader literature, indicating that excessive work demands are a primary driver of burnout in healthcare professionals [9, 14, 15]. The relentless pace, lack of adequate breaks, and the need to juggle multiple roles detract from their capacity to provide high-quality patient care and maintain personal well-being. This is not merely an issue of individual resilience; it points to systemic flaws within healthcare organisations that fail to support their staff adequately. These include inadequate support structures, inefficient work processes, and a lack of recognition of the critical need for rest and recovery among healthcare professionals [2]. By recognising and addressing these systemic issues, healthcare organisations can create a more sustainable and supportive work environment that promotes both the well-being of physiotherapists and the quality of care provided to patients.

The COVID-19 pandemic has exacerbated these challenges, introducing new stressors such as increased patient loads, the transition to remote work, and additional administrative burdens [16]. Greenberg et al. [17] elaborates on how healthcare professionals faced unprecedented pressures during the pandemic, leading to moral injury and mental health problems due to the impossible decisions they had to make, such as allocating scarce resources and balancing their own health needs with those of their patients. The authors also emphasise the importance of proactive mental health support and the implementation of routine support processes, such as peer support programs, to mitigate the negative impacts of moral injury and mental health issues. However, the issues faced by physiotherapists did not cease with the pandemic; they continued as demands increased, staff numbers reduced, and funding decreased. The lack of consistent management support during and after this period further underscores the systemic issues at play. Healthcare organisations must invest in long-term strategies to support their workforce to withstand such crises and beyond.

Perfectionism is another critical factor contributing to burnout [18]. The constant striving for perfection, while initially promoting high performance and meticulous care, eventually leads to significant stress and emotional fatigue. Our findings highlight that physiotherapists' high standards and expectations often lead to emotional exhaustion and feelings of inadequacy. Koutra et al. [19] explained that perfectionism is strongly associated with increased psychological distress. This is compounded by the lack of self-compassion, which can mediate the relationship between perfectionism and psychological distress. Self-compassion allows individuals to treat themselves with the same kindness and understanding they would offer to a friend in distress, thus mitigating the negative effects of perfectionism. Addressing perfectionism through interventions such as therapy and self-compassion practices can help mitigate its negative impact [19]. However, these individual-level interventions must be complemented by systemic changes that reduce the pressure to meet unrealistic standards such as unattainable patient quotas and expectations for perfection; proposed interventions should involve workload redistribution and time management training. Healthcare organisations must acknowledge the detrimental effects of perfectionism and take proactive steps to address it. This could include providing training, offering resources and time for therapy and mental health support, and fostering a work environment that values continuous improvement over flawless performance.

Similarly moral injury emerged as a significant issue, stemming from the conflict between professional values and the constraints of the healthcare system. This concept is well-documented in literature, where healthcare professionals face situations that force them to act against their moral or ethical beliefs due to systemic constraints [20]. Physiotherapists reported feelings of guilt, shame, and dissatisfaction, which are common symptoms of moral injury. These emotions contribute to burnout and increase the likelihood of professionals considering leaving the profession [21]. While personal coping strategies such as therapy are beneficial, healthcare organisations need to establish robust support systems that can help physiotherapists navigate the ethical dilemmas they face daily.

Team dynamics and support systems play a crucial role in mitigating stress and burnout [22]. Effective team dynamics and support from colleagues, family, and friends are essential for emotional and professional well-being. Our findings highlighted that physiotherapists face significant interpersonal challenges, such as criticism and bullying, which negatively impact their job satisfaction and mental health. The importance of supportive relationships in the workplace is well-supported by research, which shows that strong professional and personal support networks can buffer against the negative effects of job stress [23]. Social support is crucial for maintaining mental health and well-being, acting as a buffer against the stresses of the job [23]. This is particularly important in high-stress environments like healthcare, where the demands can be overwhelming. Healthcare organisations need to create work environments that encourage positive team dynamics and provides adequate support to staff of all grades. Implementing strategies to promote teamwork, open communication, and mutual respect can enhance job satisfaction and reduce burnout among physiotherapists.

The demanding nature of physiotherapy frequently results in poor work-life balance, with many professionals struggling to delineate their professional responsibilities from personal time [1, 3, 24]. This lack of balance is a significant contributor to burnout and turnover intentions. Current organisational structures often fail to support necessary flexibility in scheduling and the establishment of boundaries, exacerbating this issue. This is consistent with findings that a lack of work-life balance contributes significantly to burnout and turnover intentions among healthcare professionals [2, 25]. Fostering a culture that values and respects work-life boundaries is crucial. This could involve training for management on the importance of work-life balance, ensuring that regular breaks and time off are respected, and providing resources for mental health support. Encouraging open dialogue about the challenges of maintaining work-life balance can also help in identifying specific areas where improvements are needed. This not only enhances their well-being and job satisfaction but also improves patient care outcomes, leading to a more resilient and effective healthcare system.

## Conclusion

This study provides a comprehensive understanding of the complex relationship between workload, burnout, and perfectionism among UK physiotherapists. Addressing these issues requires systemic changes to support physiotherapists in managing their workloads and maintaining their well-being. By implementing effective policies, training, promoting professional development, and fostering supportive work environments, healthcare organisations can reduce burnout and improve the professional fulfilment and job satisfaction of physiotherapists, ultimately leading to better patient care and outcomes.

## Supporting information

**S1 File. Supplementary document on codes, categories and overarching themes formation.**
(PDF)

**S2 File. COREQ checklist.**
(PDF)

## Author Contributions

**Conceptualization:** Glykeria Skamagki, Laura Blackburn, Daniel Biggs, Cameron Black, Sivaramkumar Shanmugam.

**Data curation:** Glykeria Skamagki, Maria Kolitsida.

**Formal analysis:** Glykeria Skamagki, Maria Kolitsida.

**Investigation:** Laura Blackburn, Sivaramkumar Shanmugam.

**Methodology:** Glykeria Skamagki, Laura Blackburn, Daniel Biggs, Cameron Black, Sivaramkumar Shanmugam.

**Project administration:** Laura Blackburn, Sivaramkumar Shanmugam.

**Supervision:** Cameron Black, Sivaramkumar Shanmugam.

**Validation:** Daniel Biggs.

**Writing – original draft:** Glykeria Skamagki, Laura Blackburn, Maria Kolitsida, Cameron Black, Sivaramkumar Shanmugam.

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
