## [Decision Letter · Decision Letter 0]

15 Dec 2024

PONE-D-24-48686Exploring Burnout, Perfectionism, and Moral Injury Among UK Physiotherapists: A Qualitative Study on Professional Fulfilment and Well-beingPLOS ONE

Dear Dr. Shanmugam,

Thank you for submitting your manuscript to PLOS ONE. After careful consideration, we feel that it has merit but does not fully meet PLOS ONE’s publication criteria as it currently stands. Therefore, we invite you to submit a revised version of the manuscript that addresses the points raised during the review process.

**ACADEMIC EDITOR: **I appreciate the authors for raising this important issue of burnout. Based on the comments received from the reviewers, you are required to address it and send the revised manuscript along with enhanced language flow as per the PLOS ONE standard. I wish you all the bestPlease ensure that your decision is justified on PLOS ONE’s publication criteria and not, for example, on novelty or perceived impact.

We look forward to receiving your revised manuscript.

Kind regards,

Mohammad Sidiq, PhD Pain Sciences Physiotherapy

Academic Editor

PLOS ONE

2.  We note that your Data Availability Statement is currently as follows: [All relevant data are within the manuscript and supporting files.] Please confirm at this time whether or not your submission contains all raw data required to replicate the results of your study. Authors must share the “minimal data set” for their submission. PLOS defines the minimal data set to consist of the data required to replicate all study findings reported in the article, as well as related metadata and methods (https://journals.plos.org/plosone/s/data-availability#loc-minimal-data-set-definition).

Additional Editor Comments:

Dear Authors as per the comments recieved by the reviewers, kindly submit the revised manuscript addressing all comments raised by reviewers.

Reviewers' comments:

Reviewer's Responses to Questions

**Comments to the Author**

1. Is the manuscript technically sound, and do the data support the conclusions?

Reviewer #1: Yes

Reviewer #2: Partly

2. Has the statistical analysis been performed appropriately and rigorously? 

Reviewer #1: Yes

Reviewer #2: No

3. Have the authors made all data underlying the findings in their manuscript fully available?

Reviewer #1: Yes

Reviewer #2: Yes

4. Is the manuscript presented in an intelligible fashion and written in standard English?

Reviewer #1: Yes

Reviewer #2: Yes

5. Review Comments to the Author

Reviewer #1: Respected Author,

Greetings of the day,

To begin with, we would like to appreciate the authors for the commendable task. In continuation, we humbly request the authors to carry further the present study to diverse populations.

Regards

Reviewer #2: The sample size is very less for an exploratorive study, also the age group has not been mentioned but taken from 26 years to 50 years where burnout and injuries may vary due to aging and age related changes also they have mentioned about cancer and workplace so the reference of the theme has not been given. where as several places are not been cited

6. PLOS authors have the option to publish the peer review history of their article (what does this mean?). If published, this will include your full peer review and any attached files.

Reviewer #1: **Yes: **JYOTI SHARMA

Reviewer #2: No

---

## [Author Response · Author response to Decision Letter 0]

22 Dec 2024

Revisions 22-11-2024 - All feedback were addressed or defended in the rebuttal letter titled 'Response to Reviewers'. Also uploaded a marked up copy of the manuscript with track changes and an unmarked version of the revised paper. Also included supplementary document. Also checked an confirm the revised manuscript meets the Plos One style requirements. Data is available in the supplementary file. Reference list is reviewed and can confirm it is accurate and complete.

---

## [Editor Report · Decision Letter 1]

30 Dec 2024

Exploring Burnout, Perfectionism, and Moral Injury Among UK Physiotherapists: A Qualitative Study on Professional Fulfilment and Well-being

PONE-D-24-48686R1

Dear Dr. Shanmugam,

We’re pleased to inform you that your manuscript has been judged scientifically suitable for publication and will be formally accepted for publication once it meets all outstanding technical requirements.

Kind regards,

Mohammad Sidiq, PhD Pain Sciences Physiotherapy

Academic Editor

PLOS ONE

Additional Editor Comments (optional):

I m satisfied with the revision done by the reviewers and i recommend it to be acepted for publication.
---

## [Editor Report · Acceptance letter]

7 Jan 2025

PONE-D-24-48686R1 

PLOS ONE

Dear Dr. Shanmugam, 

I'm pleased to inform you that your manuscript has been deemed suitable for publication in PLOS ONE. Congratulations! Your manuscript is now being handed over to our production team.

Kind regards, 

on behalf of

Dr. Mohammad Sidiq 

Academic Editor

PLOS ONE